# Targeted Cellular Micropharmacies: Cells Engineered for Localized Drug Delivery

**DOI:** 10.3390/cancers12082175

**Published:** 2020-08-05

**Authors:** Thomas J. Gardner, Christopher M. Bourne, Megan M. Dacek, Keifer Kurtz, Manish Malviya, Leila Peraro, Pedro C. Silberman, Kristen C. Vogt, Mildred J. Unti, Renier Brentjens, David Scheinberg

**Affiliations:** 1Molecular Pharmacology Program, Sloan Kettering Institute, New York, NY 10065, USA; gardnert@mskcc.org (T.J.G.); bournec@mskcc.org (C.M.B.); dacekm@mskcc.org (M.M.D.); kgk4001@med.cornell.edu (K.K.); malviyam@mskcc.org (M.M.); perarol@mskcc.org (L.P.); silbermp@mskcc.org (P.C.S.); krv4001@med.cornell.edu (K.C.V.); 2Immunology Program, Weill Cornell Graduate School of Medical Sciences, New York, NY 10065, USA; 3Pharmacology Program, Weill Cornell Graduate School of Medical Sciences, New York, NY 10065, USA; mju4001@med.cornell.edu; 4Tri-Institutional PhD Program in Chemical Biology, Memorial Sloan Kettering Cancer Center, New York, NY 10065, USA; 5Department of Medicine, Memorial Hospital, New York, NY 10065, USA; brentjer@mskcc.org

**Keywords:** adoptive cell therapy, armored CARs, CAR T, chimeric antigen receptor T cell, cell engineering, gene therapy, immunotherapy, synthetic biology, synthetic immunology, targeted cellular micropharmacy, TCR therapy

## Abstract

The recent emergence of engineered cellular therapies, such as Chimeric antigen receptor (CAR) CAR T and T cell receptor (TCR) engineered T cells, has shown great promise in the treatment of various cancers. These agents aggregate and expand exponentially at the tumor site, resulting in potent immune activation and tumor clearance. Moreover, the ability to elaborate these cells with therapeutic agents, such as antibodies, enzymes, and immunostimulatory molecules, presents an unprecedented opportunity to specifically modulate the tumor microenvironment through cell-mediated drug delivery. This unique pharmacology, combined with significant advances in synthetic biology and cell engineering, has established a new paradigm for cells as vectors for drug delivery. Targeted cellular micropharmacies (TCMs) are a revolutionary new class of living drugs, which we envision will play an important role in cancer medicine and beyond. Here, we review important advances and considerations underway in developing this promising advancement in biological therapeutics.

## 1. Introduction

A central aim of modern pharmacology is to selectively treat disease while avoiding harmful effects to normal cells, tissues, and systems. This is an inherently difficult challenge as traditional therapeutic agents are distributed systemically throughout the body and can act indiscriminately. The ratio of the levels of drug needed to affect normal tissues compared to the target is known as the therapeutic index (TI) and is typically low, especially with cancer therapeutics, resulting in significant toxicity for many drugs. Monoclonal antibodies, genetic therapies, and pathway-selective agents have improved the TI for many diseases [1,2,3]. The recent emergence of engineered cells to treat cancer (in particular, chimeric antigen receptor (CAR) T cells) offers a new approach to improving the TI because these cells selectively expand exponentially in the vicinity of the target cancer cells [4,5]. Based on this unique pharmacology, new approaches to use cells to deliver drugs have arisen. Cell-mediated drug delivery, or “cellular micropharmacies”, thus represent a revolutionary approach to the challenge of more controlled and selective drug administration.

Immune cells are ideal vectors by which to selectively deliver drugs within the human body, and indeed do so during their normal physiologic functions. Certain cell types have been evolutionarily optimized to possess several pharmacokinetic properties that pharmacologists still strive to achieve in drug design, such as precise tissue localization, temporal control of action when needed, and rheostats to control activity levels locally. In summary, these cells are smarter than current systemically administered agents. As examples, lymphocytes patrol and persist within the circulatory system, selectively entering tissues and undergoing receptor-mediated clonal expansion in times of infection to constitute a local population of specialized antigen-specific cells, which then exhibit receptor-mediated shut down when not needed. The cells also secrete a number of inflammatory cytokines and other molecules as effectors in their action or to recruit other components of the immune system. As a self-constituent of the human body, they can persist and expand, and are not dispersed until finished. Finally, the cells remain in reserve as “memory cells” to re-emerge when needed.

From a structural standpoint, a human cell also possesses a number of features valuable for drug delivery. With a large volume and surface area in comparison to exosomes or nanoparticles, cells permit the expression or attachment of a large amount of therapeutic cargo. Conjugation of molecular elements via enzymatic or chemical means is also relatively facile. Furthermore, as living biological systems, human cells can be genetically altered by the introduction of customized genetic elements that enable numerous options for the expression and temporal control of delivery of biologic cargo.

Immunotherapy based on the development of cellular therapeutics has seen promising results observed with CAR T cells, T cell receptor (TCR) gene therapy, and tumor-infiltrating lymphocyte (TIL) technologies, which have garnered considerable attention for their striking efficacy in treating various malignancies. A paradigm shift in biological therapy is currently underway as various approaches emerge to co-opt these cells as carriers and producers of drugs locally to the disease site. Significant recent advances in gene editing and delivery, synthetic biology, as well as an improved understanding of the immune axes involved in cancer and disease are rapidly converging, leading to a revolution in the design and utilization of targeted cellular therapies. ‘Armored CAR T cells’, which exogenously express proteinaceous immunomodulatory agents to engender more potency or resistance of the immune cell, are among the first of these promising technologies [6]. Here, we introduce the term “targeted cellular micropharmacies” (TCM) to refer to the much broader universe of cellular therapies that have been engineered to selectively deliver therapeutic payloads of various types to a diseased tissue environment. By doing so, the TCM may boost the efficacy of existing cellular therapies, improve the therapeutic indices of the drugs, or provide entirely orthogonal pharmacologic activity through controlled delivery of therapeutic agents. We describe the important advances and considerations underway in developing this promising advancement in biological therapeutics, restricting our discussion to the use of human cells. Viruses, bacteria, exosomes, and other nanoparticles have been described elsewhere and are beyond the scope of this review [7,8,9,10].

## 2. Choice of Cells for Targeted Drug Delivery

There are various methods to obtain or generate cells that will aggregate in tissues, especially cancers. In the case of T cells, these include isolating and expanding tumor-infiltrating lymphocytes (TILs) ex vivo, or by genetically engineering a patient’s own polyclonal T cells (autologous) to express tumor-associated antigen (TAA)-specific TCRs or CARs. Human leukocyte antigen (HLA)-matched donor peripheral blood mononuclear cells (PBMC)-derived T cells (allogeneic T cells) can also be used for this purpose. Other immune cells are also capable of recognizing and eliminating cancer cells, such as Natural killer (NK) cells, NK-T cells, macrophages, and B cells. These cells use diverse trafficking, signaling, and target killing strategies that can be utilized independently or in combination with adoptive T cell therapy to eliminate immune-evading cancer cells, and can be further enhanced as TCMs to deliver therapeutic payloads to the tumor environment (Table 1).

### 2.1. Tumor-Infiltrating Lymphocytes

Adoptive cell therapy with cytokine-engineered TILs could be used to prolong the in vivo survival of transferred cells and also minimize the toxicity associated with exogenous cytokine administration [11,12]. For example, IL-2-engineered TILs displayed up to 10-fold enhanced proliferation and retention of CD8 T cells and NK cells in the tumor microenvironment (TME) with a minimal effect on Tregs [13]. Such an approach with IL-7, IL-12, IL-15, and IL-18 engineered TILs could also support TILs’ expansion and function in vivo [14,15,16,17].

### 2.2. Engineered T Cells

Through TCR gene transfer technology, a large population of polyclonal T cells can be redirected to attack and kill the specific target tumor associated antigen (TAA)-bearing cancer cells. An advantage of TCR-based targeting is that receptor affinity can be artificially enhanced up to thousands fold by substituting amino acids in the antigen-binding regions of the TCR using in vitro genetic engineering and library screening technologies [19]. TCR-engineered T cells have also shown promise in solid tumor treatment [40].

CAR-engineered T cells have revolutionized cancer therapy, with an impressive success rate in patients with B cell leukemia and lymphoma [41]. CAR T cell benefits can last for many years, with memory CAR T cells observed in various patients at long-term follow up and can be directed to a variety of cell surface targets [41,42,43,44]. However, CAR T cell therapies have not generally been successful for the majority of solid tumors due to the lack of unique TAAs, and inhibition by the immunosuppressive TME [21,45,46], with severe toxicities in some cases [47], and relapse in some patients due to either CAR T cell exhaustion [48,49] or downregulation of the cognate antigen by tumor cells [50].

### 2.3. NK Cells

Natural killer (NK) cells are large granular cytotoxic cells originating from a common lymphoid progenitor cell that detect and kill cancer cells and virally infected cells [51,52]. NK receptors can recognize both classical and non-classical human leukocyte antigen (HLA) class I molecules and transduce context-dependent activating or inhibitory signals [53,54]. Like T cells, the effector function of activated NK cells is mediated by multiple approaches, such as granzymes, perforin, Fas/FasL interaction, tumor necrosis factor (TNF)-related apoptosis-inducing ligand (TRAIL)/TRAIL receptors, FcγRIIIa (CD16a) mediated antibody-dependent cell cytotoxicity (ADCC), and the release of cytokines, such as IFNγ and TNFα, to activate and induce the homing of other innate and adaptive immune cells [55,56]. NK cells may be pre-activated with cytokines or engineered with CARs and are relatively easy to generate from umbilical cord blood, induced pluripotent stem cells (iPSCs), and the NK-92 cell line [34,57,58,59,60,61,62,63,64].

### 2.4. B Cell-Based Cancer Immunotherapy

Tumor-infiltrating B cells can be isolated from tumors and engineered to express therapeutic agents [25,26,27]. An alternative strategy to the infusion of therapeutic monoclonal antibodies is the adoptive transfer of B cells genetically engineered to secrete anti-TAA antibodies in vivo, thereby eliminating the need for multiple infusions of monoclonal antibodies [28]. Recently, genetic manipulation of B cells using CRISPR/Cas9 has been successfully demonstrated, potentially opening the door to a large number of applications for genetically engineered B cell therapies [29,30]. Furthermore, lentivirus-mediated gene transfer into human hematopoietic stem/progenitor cells (HSPCs), followed by differentiation of the transduced cells into B cells has successfully produced a broadly neutralizing anti-HIV antibody [30].

### 2.5. iPSC-Based Cancer Immunotherapy

Most immune cells, such as T cells, B cells, NK cells, DCs, and macrophages, have been successfully generated from iPSCs for cancer immunotherapy [35,36,65,66,67]. To overcome the problem of primary T cell exhaustion and senescence due to prolonged activation and proliferation, tumor antigen-specific T cell clones are reprogrammed into iPSCs (T-iPSCs) and then successfully differentiated back into T cells again (T-iPSCs T cells) [37,68,69]. Some example of such T-iPSC cells with defined specificity are anti-MART-1 (melanoma), anti-pp65 antigen (cytomegalovirus), anti-LMP2 (EBV antigen), anti-GPC3 (hepatocellular, ovarian, and lung carcinoma), and anti-WT-1 (leukemia) CD8 T cells [38,39,68,69]. Tumor antigen specificity can be assigned to iPSCs or T-iPSCs and their T cell derivatives by introducing exogenous TCR or CAR transgenes [70,71].

### 2.6. Macrophage-Based Cancer Immunotherapy

One strategy to enhance the antitumor effector function of macrophages is by adoptive transfer of macrophages genetically engineered to reduce IL-10 and PD-L1 expression, and increase the production of IL-21 and soluble TGFβ receptor [22,23]. IL-21 promotes proliferation of antitumor T cells and NK cells, whereas the soluble TGFβ receptor neutralizes immunosuppressive TGFβ by serving as a decoy receptor in the TME. Macrophages engineered to express CAR-targeting TAAs, such as anti-CD19 and anti-HER2, have recently been reported. These CAR-engineered macrophages not only sustained a proinflammatory M1 phenotype but also converted bystander immunosuppressive M2 macrophages into M1, boosted antitumor T cell activity, and demonstrated antigen-specific phagocytosis and tumor clearance [23]. Another strategy to enhance the antitumor effector function of macrophages is by blocking the CD47/SIRPα axis. Macrophages expressing an anti-CD47 CAR with an activating intracellular domain, or secreting anti-CD47 peptides/scFv may be incorporated in future therapeutic approaches [24].

## 3. Vector Design and Gene Transfer for Engineered Cells

TCMs may rely on multiple genetically encoded elements that enhance the function of the cellular therapy. Traditionally, genes are transferred to patient-derived cells via lentiviral or retroviral vector transduction, but these vectors are severely limited in size to generally about 10 and 8kb, respectively [72,73]. Therefore, various methods have been developed to enable robust expression of multiple gene elements while still maintaining a reasonable vector size (Table 2).

### 3.1. Multicistronic Vector Design

Self-cleaving peptides, such as T2A and P2A, are 18-22aa peptides derived from picornavirus that induce ribosome stalling and separate a polyprotein into two distinct cistrons [74]. Furin cleavage sites can also induce separation of a polyprotein into individual elements through cleavage by the golgi-resident protease furin [75]. A single shared promoter and the small size of these elements results in little impact on gene transfer and viral titer, while allowing the inclusion of multiple protein elements in one vector. Internal ribosome entry sites (IRESs) also permit constitutive expression of two proteins under control of the same promoter element [76,77]. Since IRES sequences are quite large (>400 bp), this is not often a desirable approach. Importantly, expression of the protein downstream of the IRES may be diminished compared to the upstream element [78], though this may be used as an approach to “titrate” the expression of the therapeutic protein [16].

### 3.2. Multiple Promoter Systems and Co-Transduction

Two or more promoters can be used to drive separate transcription of each transgene element [79]; however, careful considerations must be made so that the total vector size is kept low and promoters and terminator elements do not interfere with the expression of either transgene or the generation of infectious virus [93]. When gene constructs larger than the limit of lentiviral or retroviral vectors are needed, multiple transgene elements may be introduced via co-transduction with two or more viral preparations [80,81]. Since co-transduction requires the packaging of separate viral batches, these methods are limited due to the difficulty of large-scale viral production and the low efficiency of transducing both vectors into every cell [82].

### 3.3. Non-Viral Gene Delivery Methods

The aforementioned strategies focus on viral transduction, but gene editing technology using non-viral techniques are emerging as a promising strategy for cell editing. Zinc finger nucleases (ZFNs), TALE nucleases (TALENs), and CRISPR-Cas systems allow for targeted insertion of transgenes through homology-directed repair (HDR), which minimizes the likelihood of adverse events caused by semi-random retroviral gene insertion [94]. Current work directed toward increasing the efficiency of CRISPR-mediated gene insertion will undoubtedly lead to more sophisticated targeting and insertion of therapeutic genes [86,87,88].

Additional non-viral gene transfer methods include transposon and mRNA delivery [89]. In transposon delivery, semi-random integration of a transgene is facilitated by Sleeping Beauty or piggyBac transposon systems and co-transfection of a transposase enzyme. This method allows for larger transgene constructs but is less efficient than viral transduction and causes significant cellular toxicities [95]. mRNA-mediated transgene delivery similarly allows for larger gene cassettes, but since expression is driven by the translation of exogenously delivered mRNA, the transgene is rapidly diluted and degraded as cells divide [96].

## 4. Approaches to Engineering Cells Inside the Patient

Clinical use of viral vectors to engineer cells ex vivo, discussed briefly above, has been extensively reviewed elsewhere and includes adenoviruses, retroviruses, lentiviruses, and others [97,98,99]. While proven successful in the clinic for many B cell neoplasms, these methods are time intensive, expensive, and require unique processes for each patient. Furthermore, generating clinical-grade homogenous preparations of virus on a large scale remains a limiting factor for the scalability of adoptive cell therapies (ACTs). Developing universal technologies for editing cells in situ will help address each of these significant hurdles and expand the utility of TCMs (Table 2). Dendritic cell vaccines, reviewed elsewhere, represent another approach for artificially focusing an the antitumor response within the patient, with well-documented safety and clinical responses observed despite limited long-term clinical benefits [100,101,102].

### 4.1. Transposase Delivery

A recently developed technology that uses targeted DNA nanoparticles to deliver CAR-encoding genes specifically to T cells in situ demonstrates the feasibility of such therapies [83]. Co-delivery of piggyBac transposase and CAR plasmids allowed for the integration of CAR genes into T-cell chromosomes, permitting stable expression of leukemia-specific 19-41BBz CARs. Coupling anti-CD3e f(ab’)2 fragments to the surface of the nanocarriers enabled T-cell-specific endocytosis and cargo delivery. Importantly, T cells reprogrammed via polymeric nanocarriers demonstrated comparable efficacy to those transduced with conventional ex vivo approaches. This DNA nanocarrier technology may provide a practical alternative to traditional CAR T cell therapy, with the potential to be a broadly applicable cancer treatment.

### 4.2. mRNA Delivery

Another risk of ACT is the permanent expression of an introduced transgene, which could potentially mutate, dysregulate, disrupt an essential gene, or integrate into an incorrect cell type [5,103]. Targeted mRNA-loaded nanocarriers that reprogram T cells via transient expression could solve this problem [84]. For example, the transient expression of Foxo1 was shown to reprogram the differentiation of effector cells into functionally competent memory cells. Additionally, immunosuppressive tumor-associated macrophages were shown to be transiently reprogrammed to a proinflammatory phenotype via nanoparticles containing in vitro-transcribed mRNA encoding M1-polarizing transcription factors [85]. This cost-effective and scalable technology demonstrates the utility of transient genetic modifications in enhancing immune cell therapeutic value.

### 4.3. Cellular Implants

Implantable devices carrying a variety of biologic agents have proven useful for the site-specific delivery of engineered immune cells to sites where a tumor has been excised, or on the surface of nonresectable tumors. For example, micro-patterned nickel titanium (nitinol) films impregnated with ovarian tumor-specific CAR T cells recognizing ROR1 were shown to deliver a high density of T cells directly to solid tumors [90]. Although these biocompatible implants eradicated OVCAR-3 tumors in the treated mice, their poor biodegradability raised long-term safety concerns of permanent implantation.

Porous biopolymeric scaffolds produced from alginate, a biodegradable and biocompatible polysaccharide, present a safer alternative to nitinol films [91]. This technology has been further expanded by co-delivering engineered T cells with stimulators of IFN genes (STING) agonists, such as cyclic di-GMP, to improve the effectiveness of CAR T cell therapy [92]. These biomaterial-supported T cell implants demonstrate the utility of localized immunotherapy, especially in the treatment of solid tumors. The use of various materials to deliver transgenes in either a stable or transient manner offers a simple scalable alternative to current ex vivo viral-based transduction strategies. These innovations have the potential to expand the scope of CAR T cell therapy toward many more malignancies.

## 5. Cellular Delivery of Therapeutic Antibodies and Their Derivatives

Monoclonal antibodies and their derivatives have revolutionized cancer treatment due to their selective tumor targeting, long plasma half-life, predictable toxicities, and multiple modes of therapeutic action [104]. However, on-target off-tumor toxicities, large frequent dosing, as well as poor tumor penetration have limited their success. Most notably, checkpoint blockade antibodies against CTLA-4 and PD-1/PDL-1 are highly effective in activating immune responses against a variety of cancers but are limited by a high incidence of immune-related severe adverse events, including death [105,106,107,108]. As such, antibodies that promote immune checkpoint blockade are ideal candidates for use in cell-based local delivery of antibody therapeutics (Figure 1). This strategy can simultaneously limit the systemic toxicity of these agents to the sensitive tissues of the gastrointestinal tract, skin, and endocrine organs, while aiding in local antibody penetration within the tumor milieu.

### 5.1. CAR T Cells Secreting Antibodies to PDL-1/PD-1

CAR T cell activity is inhibited by overexpression of PDL-1 and PDL-2 on tumor cells [109,110], as well as by expression of PD-1 on the CAR T cells themselves [110,111,112,113]. Accordingly, in a model of renal cell carcinoma (RCC), CAR T cells targeting carbonic anhydrase IX were engineered to secrete full-length anti-PDL-1 antibodies [114]. Enhanced killing of PDL-1-expressing tumors in vitro and in vivo was observed, as well as a decrease in T cell surface exhaustion markers PD-1, Tim3, and Lag3 on TILs. The-full length IgG1 was able to recruit NK cells in vivo, as well as induce antibody dependent cellular cytotoxicity (ADCC) in vitro [114]. Similarly, enhanced efficacy was seen in a fully immune competent model with CAR T cells secreting anti-PD-1 single chain variable fragments (scFvs) [110,115]. Strikingly, CAR T cells secreting PD-1 scFv persisted in surviving mice and re-invigorated endogenous TILs were observed upon re-challenge [110].

### 5.2. CAR T Cells Secreting Antibodies to CTLA4

CTLA-4 knockout in T cells [116] and knockdown in CAR T cells [117] leads to enhanced cytolytic efficacy of these effector cells. Consequently, CAR T cells secreting anti-CTLA-4 antibody fragments are being developed for clinical use. Interestingly, in a model of glioma, IL13-Ra2-targeted CAR T cells engineered to secrete an anti-CTLA-4 minibody, but not anti-PD1 or –TIM3, were able to significantly control tumor growth [118]. In addition, CARs targeting EIIIB modified to secrete anti-CTLA-4 single-domain antibodies showed decreased expression of PD-1 and PDL-1 on the CAR T cells and improved persistence in an immune-competent tumor model [119].

### 5.3. Cells that Disrupt the CD47–SIRPα Signaling Axis

Blockade of the CD47–SIRPα signaling axis, often referred to as the “do not eat me” signal [120,121] through cell secretion, improved FcγR-mediated phagocytosis and cross presentation by dendritic cells [122,123,124,125]. CAR T cells secreting single-domain antibody fragments (VHHs) against CD47 showed improved antitumor activity and epitope spreading in syngeneic models when combined with the TA99 antibody against B16-F10 melanoma cells [126]. In an alternative approach, CAR T cells secreting CV1, a high-affinity peptide that also disrupts the CD47–SIRPα axis, also improved efficacy in mouse models of lymphoma [127].

### 5.4. Cellular Delivery of Antibodies Against Tumor-Associated Antigens

Local antibody secretion can be used to improve specificity against TAAs [104], such as Her2 and EGFR, which can be overexpressed by cancers but are also found on a variety of healthy tissues, leading to toxicities when targeted by systemic antibody-based therapeutics [128,129,130]. To overcome this limitation for the EGFR variant, CAR T cells directed toward EGFRvIII were engineered to secrete a bispecific T cell engager (BiTE) against wild-type EGFR, found on the larger proportion of glioblastoma cells. BiTEs secreted locally in glioblastoma tissue were able to redirect non-CAR T cells to kill the tumor, resulting in superior tumor clearance and mitigation of antigen loss from the tumor mass [131].

## 6. Delivery of Cellular-Modulating Agents in Cancer

The TME is often comprised of inhibitory factors that blunt the immune response, including cellular elements, such as myeloid-derived suppressor cells, regulatory T cells (Tregs), and tumor-associated macrophages. ACT therapies, such as CAR T cells, must overcome these barriers in order to effectively manage the full breadth of cancer types. TCMs offer a promising strategy to deliver elements to the TME that may potentiate the antitumor immune response, such as with armored CAR T cells, or provide orthogonal antitumor activity by sensitizing tumor cells to immune clearance (Figure 1).

### 6.1. TCMs Expressing Tumor Suppressor Proteins

Loss-of-function mutations in tumor suppressor genes and their regulators can result in the onset and progression of cancers [132]. Continuous delivery of functional tumor suppressor protein to the tumor by cellular micropharmacies can overcome these mutations. For example, herpes virus entry mediator (HVEM) is frequently mutated in germinal center lymphomas, stimulating neoplastic B cell receptor signaling and B cell growth. CD19-targeted CAR T cells engineered to deliver soluble HVEM to lymphomas showed superior efficacy than CAR T cells alone due to soluble HVEM binding to the inhibitory receptor BTLA on B cells [133]. This creative approach highlights the potential role of TCMs as therapeutic agents that can directly deliver an unstable protein or peptide to the diseased tissue.

TCMs can be genetically programmed to carry diverse agents into the tumor environment. These elements may potentiate the activity of the TCM itself, condition the tumor microenvironment by activating additional immune effector cells, or provide an orthogonal antitumor activity.

### 6.2. Cytokines in the TME

Immune effector proteins, such as cytokines and costimulatory proteins, act on tissues to orchestrate local immune responses [134,135,136,137]. However, excessive systemic elevation in cytokine levels can be toxic or fatal [138,139]. Furthermore, suppressive cytokines may support tumor growth by inhibiting antitumor immune effector cells. Therefore, altering the balance of stimulatory effector proteins specifically in the tumor milieu represents an important opportunity for therapeutic intervention. While the direct cytotoxicity of tumor-specific lymphocytes requires engagement of the cognate antigen [140,141], cytokine secretion and costimulation can act non-specifically to promote inflammation through the release of proinflammatory type 1 cytokines by other cells in the tumor [134,142,143]. Efforts to augment cytokine delivery by adoptively administered lymphocytes has achieved success in both hematologic and solid tumors.

### 6.3. Cytokines that Promote T Cell Persistence

Early attempts to create cellular therapeutic agents that secrete immune-modulating proteins began with cytokines that support T cell proliferation and persistence, including IL-2 and IL-15, using TILs from patients. TILs engineered to express IL-2 had increased persistence in vitro but failed to demonstrate efficacy or enhanced persistence in melanoma patients [12], possibly due to the promotion of regulatory T cells (Tregs) [144]. IL-15 promotes persistence and memory in T cells without Treg activation [145], leading investigators to engineer adoptive T cells that secrete exogenous IL-15. These T cells persisted without the need for exogenous cytokine in vitro and persisted significantly better than wild-type T cells in mice [17,146]. To address the risk of possible leukemic transformation or unrestrained proliferation, a kill switch was also encoded in IL-15 CAR T cells [147]. These studies indicated that rational genetic engineering can tune the efficacy and safety profiles of cellular therapeutic agents. Similar findings were seen with NK cells, which can have antileukemic effects without antigen specificity [148].

More recent efforts to engineer cytokine-enhanced tumor-specific T cells have attempted to engage endogenous effector cells with secreted CCL19, CCL21, and IL-7. CAR T cells that produce CCL19 and IL-7 turned immunosuppressive tumors into T cell niches by recruiting endogenous T cells to tumors. Interestingly these modified CAR T cells performed best without preconditioning, suggesting an important role for endogenous T cells in an antitumor response [149].

IL-23 has effects on Th1 and type 3, Th17 antitumor responses. CAR T cells express the shared p19 subunit of IL-12 and IL-23. Therefore, engineering cells to constitutively secrete p40, the second subunit of IL-23, led to activation-inducible IL-23-secreting cells that resisted exhaustion, were less toxic, and performed better in neuroblastoma and pancreatic cancer models [150].

### 6.4. TCMs that Prime Immune Effectors

Priming or costimulation is an essential component of T cell activation. Inclusion of costimulatory domains in CAR constructs, such as CD80 and 4-1BBL, markedly improved CAR T cell efficacy [151] and can also prime bystander T cells by trans-activation and auto-activation of T cells [152]. A similar approach was taken by overexpressing CD40L in CAR T cells with a potential added advantage of activation of innate antitumor immunity [153]. CD40L-expressing CARs upregulated antigen presentation in B cell malignancies and antigen-presenting cells in syngeneic mouse models of B cell lymphoma [154].

IL-12 is secreted by professional antigen-presenting cells and activates IFN-y secretion in T cells and NK cells, but it has proved too toxic for systemic use. Fibroblasts, dendritic cells, and tumor cells themselves have been engineered as IL-12 micropharmacies. Fibroblasts engineered to secrete IL-12 successfully eradicated established solid tumors when injected directly into the tumor, and T-cell-mediated abscopal effects were observed [155]. In patients, this therapeutic modality demonstrated modest clinical efficacy, and activated endogenous T cells [156]. Engineered DCs that secrete both IL-12 and IL-18 were able to eradicate established solid tumors and also produced abscopal effects when given intratumorally in mice [157]. Nonetheless, IL-12-secreting DCs had modest antitumor effects in patients [156]. Because intratumoral injections are not feasible for a majority of tumors, T cells, which are capable of homing and accumulating in tumors, have been successfully used as TCMs to deliver IL12, demonstrating potent antitumor effects in syngeneic melanoma, lymphoma, and ovarian cancer models [158]. Additionally, T cell-secreted IL-12 reprogrammed innate immune cells to adopt an inflammatory type 1 immune response [6,16,158,159,160,161]. IL-18 is secreted in conjunction with IL-12, and these cytokines synergize to stimulate IFN-y production in T cells and NK cells, providing potent antitumor responses in pancreatic and lung cancer models. IL-18 secretion also polarized T cells to memory differentiation while skewing the tumor microenvironment towards an inflammatory state [162]. Recently, T cells have been engineered to secrete Flt3L, a potent dendritic cell growth factor. These Flt3L-armored T cells promoted DC development and epitope spreading [163]. Granulocyte-macrophage colony-stimulating factor (GMCSF) cellular immunotherapy (GVAX) has also been explored for recruitment and activation of DCs to boost the antitumor response [164]

While cytokines orchestrate immune responses, microbial products are also potent inflammatory agents [165]. An interesting TCM approach could convert “cold” tumors to “hot” by the expression of pathogen-associated molecular patterns (PAMPS), such as flagellin, which are constituents of microbial pathogens that could also be delivered by T cell micropharmacies [166].

## 7. Enzyme Delivery Strategies

An alternative cell engineering approach takes advantage of the ability of TCMs to express and deliver exogenous enzymes to the tumor microenvironment to add new functionality. One such approach used solid tumor-targeting CAR T cells that express the enzyme heparanase (HPSE), which degrades heparin sulfate proteoglycans, a major constituent of the extracellular matrix [167]. Long-term ex vivo expanded CAR T cells engineered to express HPSE degraded ECM at greater levels, which promoted T cell infiltration and antitumor activity in a matrigel-embedded neuroblastoma tumor model. Another strategy, which addresses the generation of reactive oxygen species (ROS) in the tumor microenvironment, uses CAR T cells engineered to express a catalase enzyme [168].

A TCM platform recently developed by our lab enables an adoptively transferred cell to effectively synthesize a small molecule at the tumor site [169,170]. The strategy, called synthetic enzyme armed killer cells (SEAKER), utilizes TCM-mediated delivery of prodrug-activating enzymes to the tumor. Subsequent administration of a non-toxic prodrug results in conversion to a potent chemotherapeutic agent only when it encounters the enzyme, providing a localized antitumor effect that vastly improves drug TI [171]. In this TCM approach, CAR T or transgenic TCR cells are engineered to express one of two bacterial enzymes, *pseudomonas* carboxypeptidase or *enterobacter* beta-lactamase. These enzymes are robustly secreted at sites of SEAKER cell aggregation and expansion. Systemic delivery of any one of a panel of prodrugs results in localized chemotherapy activation and enhanced tumor clearance. Analogous enzyme prodrug strategies have been developed in macrophage-based platforms [172,173]. Importantly, the clonal expansion of activated SEAKER cells, together with the processivity of the exogenous enzymes, leads to a significant amplification of prodrug activation at the site of the tumor, and couples the antitumor effect of an adoptively transferred cell to potent localized chemotherapy.

## 8. Cellular Gating Strategies

Current engineered cell therapies for the treatment of cancer are typically limited by a lack of truly tumor-specific antigens, leading to adverse effects on normal cells and tissues. To address this challenge, “gating” strategies are under development to improve the selectivity of current generation cellular therapies, primarily CAR T-cells. These expression systems are activated by simple circuits that lead to a logic gate function (i.e., if cells see A, then they do B; if cells see A and B, then they do C; and so on.) The benefit of gating controls is particularly important for the development of TCMs that express additional agents to enhance their potency. If not properly controlled, aberrant activation or secretion of therapeutic cargo may pose an even greater risk to patients. In addition, as engineered cells may not remain completely localized to the tumor site, restricting the delivery of adjuvant technologies only to where and when the cell encounters its targets should be beneficial by use of smarter logic-gated systems (Figure 2 and Table 3).

Gating strategies provide a sophisticated way to modulate the expression of therapeutic cargo delivered by TCMs. Current technologies include autonomous systems, which are programmed to automatically respond to an external stimulus, or remote-controlled systems, which allow toggled expression or localization of the cellular agent or cargo.

### 8.1. Autonomous Gating Systems

Current FDA-approved CAR T cells are designed to kill cells based solely on tumor antigen recognition, such as “if antigen, then activate to kill cell”. This simple autonomous logic gate can cause adverse events, such as “on target, off tumor” killing [192]. Gated autonomous systems attempt to address this limitation by introducing more sophisticated instruction sets for cells in order to kill.

### 8.2. Activation-Dependent Systems

CAR T cells can be designed to activate upon CAR binding to two or more different antigens on a target cell by distributing costimulatory domains between multiple CARs, requiring recognition of both antigens to activate, which may be important when designing engineered T cell therapies to express products that are systemically toxic, such as in TCMs [175]. Multiantigen systems are an “if multiple antigens, then activate to kill cells” logic gate.

Activation via canonical T cell stimulatory domains, such as CD28 and 4-1BB, drives various transcription programs, including genes downstream of the nuclear factor of activated T cells (NFAT) promotor. NFAT promotors are thus the most commonly used system for inducible production of additional agents in CAR T cell therapies because proteins driven under NFAT activation require CAR T cell activation [176]. Activation-dependent systems are an “if activated, then initiate transcription of transgene” logic gate.

### 8.3. Activation-Independent Systems

Decoupling cellular output from CAR activation allows for independent cellular production of cytokines and/or therapeutic cargo. Synthetic Notch (synNotch) receptor systems enable control of cellular output upon engagement of the synNotch receptor with membrane-bound ligands on target cells [166,177,178]. In this system, both the synNotch receptor and the cellular response can be customized. SynNotch systems use a cell surface synNotch receptor comprised of an scFv that binds to the antigen of choice, a transmembrane domain adapted from the endogenous Notch receptor, and a membrane-sequestered intracellular transcription factor. Following synNotch receptor binding to its target ligand, the transmembrane region is cleaved, and intracellular portions of the synNotch receptor translocate to the nucleus and initiate transcription at its promoter. SynNotch-induced expression can drive virtually any class of biologic agent, including additional CARs, cytokines, proteins, and enzymes. SynNotch systems are an “if membrane-bound ligand, then initiate transcription of transgene” logic gate.

Modular extracellular sensor architecture (MESA) receptors enable control of cellular output upon engagement with a soluble antigen [179]. MESA receptors utilize two distinct scFvs targeting the same antigen. When both MESA receptors bind to the target antigen, their close proximity causes the protease from one MESA receptor to release the membrane-bound transcription factor from the other MESA receptor, which initiates transcription downstream from its promotor. MESA receptor technology has been used to express CARs but can be designed to drive expression of any genetically programmable biologic. MESA receptor systems are an “if soluble ligand, then initiate transcription of transgene” logic gate.

Solid tumors often grow within TMEs that can be exploited by CAR T cells. Often, TMEs are hypoxic compared to normal tissue, which has prompted the development of hypoxia-induced gating systems to drive CAR expression [180]. Using the Hif1alpha motif on CAR proteins, HIF-CARs are constitutively degraded under normoxic conditions (21% oxygen) and therefore have limited CAR surface expression. Under hypoxic conditions (1%), HIF-CAR protein degradation is reduced, therefore enabling therapeutic levels of HIF-CAR surface expression and CAR T cell tumor killing. Alternative strategies that take advantage of TME-associated proteases have also been developed. CARs in these systems are unable to bind to their cognate antigen due to the presence of small cleavable peptides that block their function. Upon cleavage by TME-associated proteases, the CARs are unmasked, and TAA-specific killing is enabled [181]. These systems are “if TME, then do B” logic gates, which may be incorporated into cellular micropharmacies.

iCARs are inhibitory receptors used for the inhibition of T cell activation, which consist of an extracellular scFv linked to the intracellular signaling domains of PD1 or CTLA4 inhibitory receptors [182]. In contrast to traditional CAR signaling, T cell activation is inhibited upon binding of an iCAR to its ligand. By co-opting these signaling pathways, constitutive production of cytokines and enzymes, or other cell functions can be inhibited. iCARs are an “if off-target ligand, then do not kill cell antigen-positive cells” logic gate.

### 8.4. Remote-Controlled Gating Systems

While autonomous systems have inherent limitations due to their lack of user control, remote-controlled systems allow direct control of CAR T cell function and can permit pharmacological regulation of TCMs.

Inducible caspase-9 (iCasp9) and anti-EGFRt antibody are systems developed as kill switches for CAR T cell therapy [183,184]. In kill switch systems, CAR T cells are transduced with transgenes encoded to respond to exogenous administration of selective drugs that will kill CAR T cells to end treatment in the case of toxicity. These systems are an “if drug is present, then end therapy” logic gate.

SUPRACARs and UniCARs are universal CAR systems that direct a generic CAR T cell to specific targets [185,186]. Unlike traditional CARs, antigen engagement is mediated through binding of a bispecific antibody directed toward a tumor antigen. Upon infusion, the bispecific antibody engages the SUPRACAR/UniCAR T cell and the tumor cell, initiating T cell activation. These modular therapies can be designed to target multiple antigens at once without the need to engineer new CARs. Furthermore, the activity of the CAR T cells is tunable based on the affinity and administration of the bispecific antibody. SUPRA and UniCAR systems are an “if modular recognition molecule bound to cells, then kill cells” logic gate.

Synthetic receptor–ligand strategies permit an engineered cell to respond independently of endogenous signaling molecules. Mutated IL2 receptor and IL2 ligands have allowed investigators to control T cell expansion, survival, and function independent of endogenous IL2 signaling [187]. Synthetic ligand and substrate pairs have also been used to create inducible CAR systems controlled by the administration of a drug through drug-on and drug-off systems [188,189]. These tools allow for exogenous drug control of CAR T cells and can be used in cellular micropharmacy technologies. Synthetic receptor–ligand pairs are an “if drug, then do or do not do B” logic gate.

Geography-restricted gating technologies enable inducible CAR expression and directed cell trafficking based on user-determined location. The light-inducible nuclear translocation and dimerization (LINTAD) system uses optogenetics to express CAR upon the administration of light [191]. LINTAD modulates the expression of CARs through a user-controlled light source in shallow tissues for the treatment of diseases, such as melanoma in the skin. Alternatively, CAR T cells can be magnetized ex vivo by membrane-coated or internalized magnetic particles. External magnetic fields can subsequently guide them to the tumor site [190]. Geo-restricted gating technologies are an “if localized external stimulus, then kill do B” logic gate.

### 8.5. Challenges of Gated Systems in TCMs

TCMs are complex systems that require careful assessment of cytokine and enzyme production to ensure reproducibility and safety. When constitutively expressed, cytokine and enzyme levels can be measured and monitored, and CAR T cell therapy can be dosed accordingly. However, in gated systems, the amount of therapeutic fluctuates with the gating stimulus. While providing an extra level of control, it also adds a layer of complexity when considering the therapeutic window for cellular therapies, especially in autonomous systems where control is limited. Remote gating through ex vivo stimuli could help maintain armored T cells and micropharmacies within the therapeutic window by controlling the dosage as needed.

## 9. Non-Genetic Engineered TCs

While the genetic reprogramming of cells described above is a powerful tool [193], gene transfer has several important limitations that have precluded their widespread clinical implementation. Non-genetic engineering strategies, which are less laborious than viral transduction, allow for the controlled ex vivo modification of cells that does not typically impact intracellular signaling or the cellular phenotype. Additionally, such alterations are transient processes that allow timely reversal of the cellular modification to prevent unwanted side effects and toxicities. A key benefit of non-genetic engineering is that its modifications can include both the introduction of biological agents (proteins, antibodies, and peptides) as well as elements that cannot be genetically programmed into a cell (small molecules, oligonucleotides, polymers, nanoparticles, and more) [194]. Prior reviews have described in detail how cells can be utilized as drug delivery agents [195,196] and how nanotechnology can enhance immunotherapy [197]. Here, we will focus on the central approaches to non-genetic engineering of cellular therapies, which can be categorized into three major groups: Encapsulation of molecules inside cells, non-covalent cell surface modification, and covalent cell membrane conjugation (Figure 3).

Cells may be functionalized to carry therapeutic cargo through non-genetic means. These include (1) encapsulation through osmosis, phagocytosis, or endocytosis; (2) non-covalent modifications through hydrophobic insertion into the lipid bilayer, liposome-mediated membrane fusion resulting in cell surface decoration, or coating the cell with positively charged particles; and (3) covalent modifications, including amine/thiol chemistry, metabolic engineering resulting in modified surface residues that enable coupling, or biotinylation of the cell surface followed by attachment of streptavidin fusion molecules.

### 9.1. Intracellular Encapsulation

Small molecules, peptides, and enzymes can be readily encapsulated into erythrocytes with osmotic pressure to improve drug pharmacokinetics [198,199,200]. Erythrocytes encapsulating L-asparaginase are currently being tested in a phase II trial for the treatment of Philadelphia chromosome-negative acute lymphoblastic leukemia [199]. Autologous transfer of erythrocytes encapsulated with dexamethasone sodium phosphate allowed for slower release of the drug over a period of 20–30 days, and showed a significant improvement of neurological symptoms in a phase II clinical trial for ataxia–telangiectasia [198].

The phagocytic properties of macrophages can also be leveraged for drug encapsulation to improve biodistribution. Such applications include the delivery of Au nanoshells to inaccessible solid tumors [201], and loading of the nanoparticle-encapsulated HIV-1 drug Indinavir (NP-IDV) [202]. Additional circulating leukocytes have also been loaded with nanoparticles as delivery vehicles for the treatment of inflammation and cancers [203].

### 9.2. Non-Covalent Surface Modifications

Modifications that rely on physical and electrostatic interactions as well as hydrophobic insertion and liposome fusion with the cell membrane are amenable to any cell type and have minimal impact on cell function. Cellular “hitchhiking” relies on electrostatic interactions, where retroviral particles adhere to the cellular membrane and are used to systemically deliver therapeutic genes. Hitchhiking viruses encoding IL-12 or herpes simplex virus thymidine kinase (HSVtk) enhanced the immunotherapeutic effect of OT-1 T cells in mice [204].

Hydrophobic insertion of chemical moieties into the cell membrane is an appealing and noninvasive method of cell surface modification. Molecules conjugated to alkyl chains, polymers, aptamers, nucleotides, lipids, and glycolphosphatidylinositol (GPI) anchors have been embedded in the cell membrane [205]. An alternative approach, which yields hydrophobic incorporation into the membrane, is liposome fusion, where phospholipid vesicles possessing unique chemical moieties fuse with the cell membrane and decorate the entire cell surface [206]. Such an approach could be applied to display targeting antibodies or small molecule moieties.

### 9.3. Covalent Membrane Conjugations

The most straightforward method for covalent conjugation to the cell membrane is via primary amines or thiols [207,208]. In a phase 1 clinical trial for the treatment of multiple sclerosis, patients were infused with autologous PBMCs decorated with myelin-derived peptides conjugated to the surface using amine chemistry in order to achieve antigen-specific tolerance [209]. Direct conjugation has also been widely used to biotinylate the cell surface [210], followed by “bridging” to streptavidin-fusion molecules to attach cargo (enzymes, antibodies, carbohydrates) to the cell [211,212].

More complex surface labeling can also be achieved by converting native surface groups into amine-reactive residues [213], as well as through metabolic engineering to incorporate unnatural amino sugar analogs [214]. NK cells metabolically engineered to display a sialic acid derivative showed enhanced binding to CD22-positive tumor cells, resulting in tumor killing and therapeutic efficacy [215].

In cellular “backpacking”, synthetic nanoparticles loaded with drug cargo are attached to the cell to provide cytokine support for tumor-targeting T cells [216]. This strategy can also be used to deliver a potent chemotherapeutic agent by loading it on T cells or NK cells that traffic to tumor sites [208,217,218,219]. Backpacks reversibly coupled to the cell surface via disulfide bonds between thiols and cargo-loaded nanoparticles allow for triggered release upon the increased surface reduction potential following TCR activation [217,218]. Disulfide-linked IL-15 nanogel backpacks were used to improve the therapeutic activity of T cells by increasing their intratumoral expansion and allowing increased cytokine dosing [217]. CAR T cells cross-linked with liposomes containing A2a adenosine receptor antagonist displayed enhanced efficacy in an ovarian cancer xenograft model and rescued hypofunctional TILs in the tumors [220].

## 10. Additional Considerations and Applications in the Clinical Use of TCMs

As the technologies discussed above advance, the opportunities for TCM therapies will greatly increase. The primary determinants for the selection of an appropriate TCM will include the nature of the desired tissue and disease target, acuteness of the disease, as well as potential resistance mechanisms. Cellular localization of the ligand targeted by both the cell and the delivered cargo (membrane bound, cytosolic, or nuclear target), relative abundance and distribution of the targets, and pharmacokinetics of the cells and the cargo as well as their mechanisms of action are also critical considerations (Table 4). Despite the evolution of these therapies, TCM strategies may not be plausible for the treatment of various forms of cancers where tumor infiltration by the TCM is not possible, or potential off-target effects are too harmful. For those cancers where TCMs do offer an advantage, various considerations must be taken to maximize clinical efficacy.

Pharmacokinetic and pharmacodynamic (PK/PD) relationships of the delivered cargo must be evaluated for the use of TCMs. As precise temporal control of the cargo is not currently feasible after cell administration, targets that require specific time-dependent target modulation may be difficult to treat with current technology. However, the PK/PD profile of TCMs and their cargo can be altered as more advanced cell engineering and gating technologies are developed (Figure 4). For example, transient delivery of TCMs traditionally results in a C_max_-dependent PK/PD profile. As such, targets that respond optimally to this PK/PD profile would be best suited for this infusion method. Activation-induced cargo release, logic gates, feedback loops, and remote-control strategies will likely permit a more sophisticated PK/PD profile that can be tailored to the specific disease and tissue. Such features may be particularly important for the treatment of advanced cancer progression, such as disseminated metastatic disease, or instances where tumor escape limits the efficacy of a single agent. Notably, due to the delay in achieving cellular localization at the disease site after cell infusion, the therapeutic benefit of the TCM may not be useful in urgent situations. For example, the use of CAR T cells targeting the fungal cell wall protein β-glucan has been proposed as a potential tool in combating *Aspergillus* infections [221], but this would be limited to patients in which immediate treatment is not critical. Long-term treatment, on the other hand, may present its own limitations, such as potential immunogenicity of the engineered cellular agent, or transformation and outgrowth of a TCM. The growing number of suicide switch strategies for deactivating engineered cells will likely address many of these concerns.

Current TCMs possess transient drug delivery profiles that wane as the adoptively transferred cells are cleared from circulation. Future technologies will permit extended drug delivery profiles, such as with cellular implants or memory T cells, oscillating drug delivery enabled through feedback loops, or pulsed drug delivery mediated by remote-controlled gating systems.

### 10.1. Mesenchymal Stem Cell (MSC)-Based Strategies for Cancer Treatment

As discussed above, a wide range of cell types have been engineered as micropharmacy treatments for cancer, including CAR T, natural killer (NK), and macrophages. MSCs represent an additional micropharmacy platform for controlled release of therapeutic agents. Preclinical studies of an MSC-delivered enzyme prodrug using thymidine kinase/ganciclovir and cytosine deaminase/5-FU enzyme prodrug systems have been shown to be effective in the treatment of glioblastoma, melanoma, gastric, hepatic, and pancreatic cancers [222]. In addition to enzyme prodrug therapy, MSCs transduced with anticancer protein ligands, such as TRAIL, TNF-α, and NK-4, and immunomodulatory cytokines, such as IL-2 and IL-12, increase the survival rates of tumor-bearing rodents [223].

### 10.2. TCM for Autoimmune Disease

MSCs, dendritic cells, and T-cells have been engineered as micropharmacies to promote immunosuppression in preclinical models of autoimmune disease. In a collagen-induced mouse model of arthritis, TGF-β-producing MSCs prevented joint inflammation via the promotion of Treg expansion and inhibition of T_H_17 cell production [224]. Additionally, TGF-β MSCs reduced the severity of acute graft versus host disease attributed to a shift in the MSC-derived macrophage population to a predominately anti-inflammatory M2 subtype [225]. Besides promoting specific differentiation patterns, MSCs have been proposed for use in enzyme prodrug therapy to selectively ablate rapidly proliferating keratinocytes that contribute to psoriasis, a skin disorder caused by excessive proliferation of cells in the epidermis [9]. Tolerogenic dendritic cells (tDCs) engineered to produce high levels of vitamin A and D metabolites have been utilized as an immunosuppressive therapy in mouse models of TNBS-induced colitis. Artificial tDC synthesis of vitamin A and D metabolites was shown to promote T cell migration to the gut and induce Treg expansion, respectively [226].

### 10.3. TCM for Neurological Disorders

Spinal cord injury repair is one well-studied example of a TCM strategy. Brain-derived neurotrophic factor (BDNF) has been delivered to injured spinal cord sites with cells to enhance neurite growth [227]. Synthetic BDNF-secreting Schwann cells and fibroblasts are two cell types that have successfully restored axons in rat models of spinal cord injury. Macrophages have also been used to treat neurodegeneration due to their ability to deliver therapeutic proteins, such as glial cell line-derived neurotrophic factor (GDNF), and promote cellular regeneration via M2-induced mechanisms [228].

### 10.4. Other Disease Targets of TCM

Acute lung conditions, bone regeneration, and fibrosis represent additional targets of TCMs. MSCs expressing exogenous IL-10 prevent ischemia reperfusion injury, a common side effect of lung allotransplantation, in rats through a reduction of the lung inflammatory response [229]. Additionally, several animal models of acute respiratory distress syndrome (ARDS) have been successfully treated with MSCs or fibroblasts modified to express angiopoietin 1, a protein that antagonizes vascular inflammation observed in ARDS [230]. MSCs have also been equipped with BMP-2, a bone morphogenetic protein that is FDA approved for the treatment of skeletal defects, to promote bone growth in mouse models simulating radial defects [231]. Several cell engineering strategies have been utilized to treat fibrosis. For example, MSCs transduced with microRNA-let7c, a negative regulator of the pro-fibrotic TGF-β, reduce kidney fibrosis in mouse models of unilateral ureteral obstruction [232]. In chronic liver fibrosis models, MSCs engineered to produce insulin-like growth factor 1 reduced fibrotic pathology, collagen deposition, and pro-fibrotic protein expression [233].

## 11. Conclusions/Perspectives

TCMs are a revolutionary class of agents that embody critical advances in modern synthetic biology, in which cellular engineering, chemistry, and immunology are applied to achieve an unprecedented level of diversity and control of drug delivery. Continued rapid innovation in these areas is accelerating more of these products into the clinic. While the development of these technologies requires a daunting blend of expertise, the clinical success already observed for armored CAR T cells points to the promise of these next-generation cell-based strategies. Investigators have now turned their attention to broadening the therapeutic benefit of cellular therapies, not only beyond cancer, but also to uncovering myriad strategies to improve the efficacy and scalability of these therapies. These efforts have shed light on the vast potential for clever and more complex engineering. At the current time, most of the technologies discussed in this review are not available as FDA-approved therapeutic agents. To achieve the broad implementation and success of TCMs, researchers will need to develop strategies to simplify or lower the cost of cell engineering, either through the use of off-the-shelf technologies, in vivo engineering, or more efficient methods of gene delivery and cell transfer. Improvements in the immunogenicity and safety profile of these products by the use of de-immunization, kill switches, logic gates, and remote control will permit more sophisticated control of their activity and greater therapeutic indices. Critical to the advancement of TCMs will be a greater understanding of the biodistribution and pharmacokinetics of adoptively transferred cells, and how the addition of therapeutic cargo may modulate those properties.

We are at the beginning of a revolution in the treatment of serious human illnesses, much as we were two decades ago with the introduction of monoclonal antibodies, which was a new class of drugs at that time, and now dominates the treatment of many cancers and autoimmune disorders. We expect that 10–20 years hence, targeted smart cellular micropharmacies will hold a prominent role in the treatment of many human diseases.

## Figures and Tables

**Figure 1 cancers-12-02175-f001:**
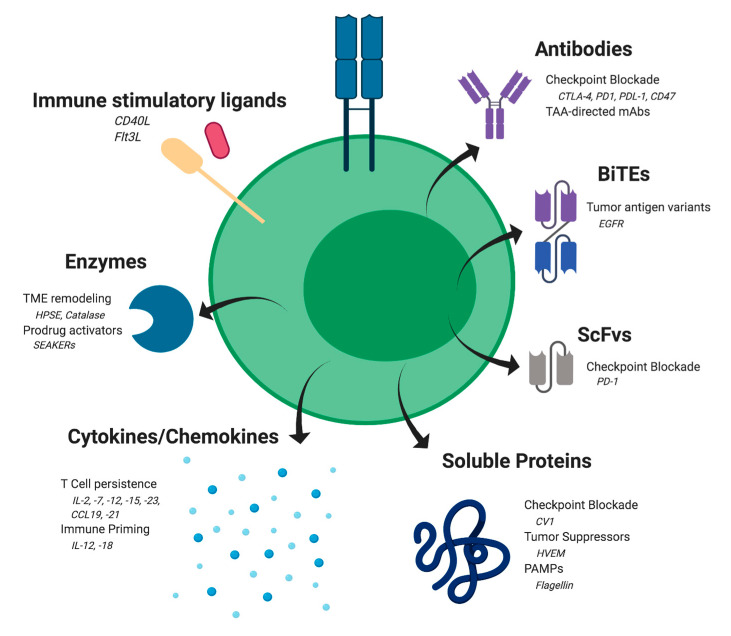
Therapeutic cargo delivered by TCMs.

**Figure 2 cancers-12-02175-f002:**
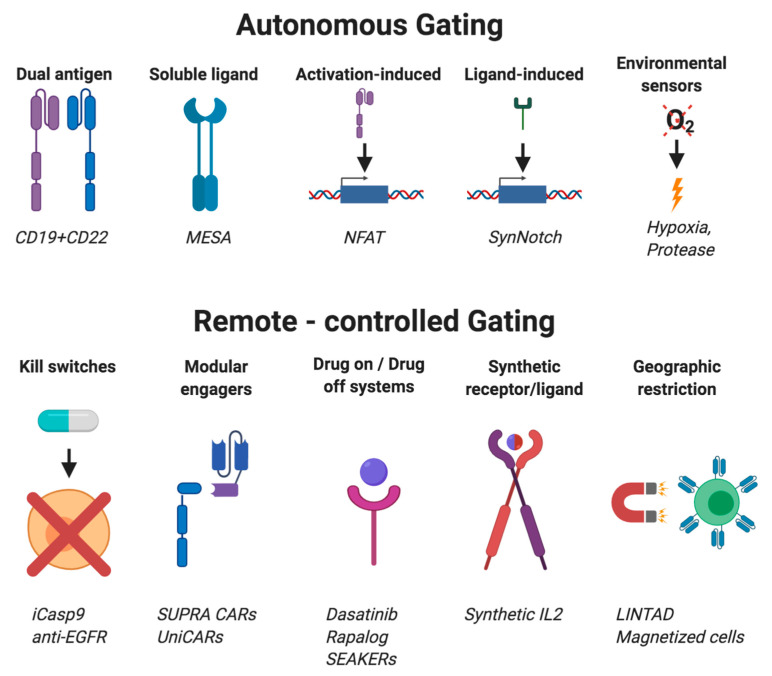
Gating strategies utilized by TCMs.

**Figure 3 cancers-12-02175-f003:**
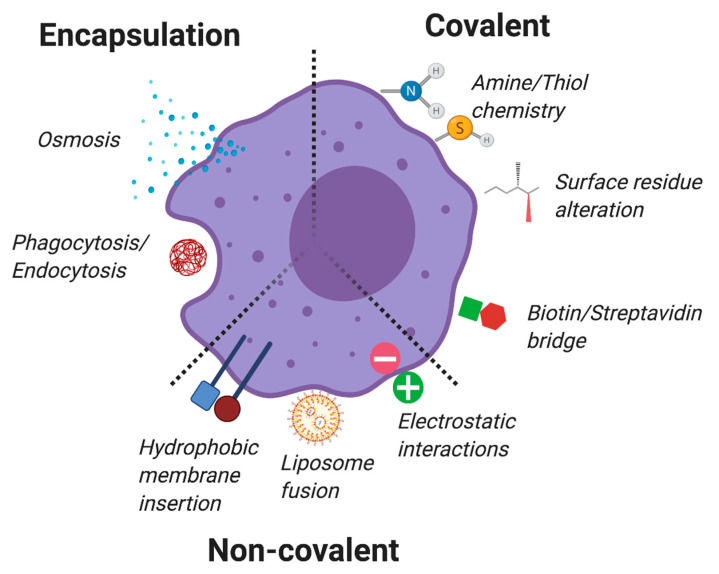
Non-genetic engineering of TCMs.

**Figure 4 cancers-12-02175-f004:**
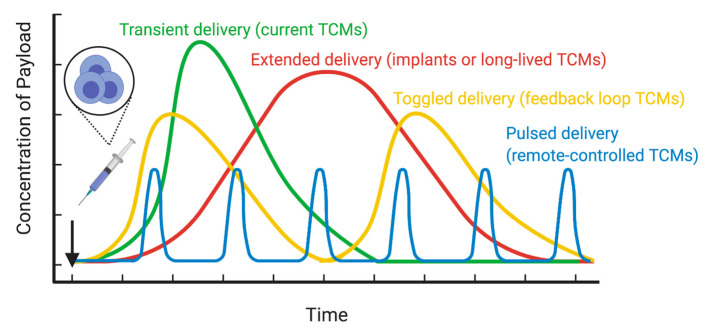
Hypothetical TCM drug delivery profiles.

**Table 1 cancers-12-02175-t001:** Choices of cells in which to engineer a TCM.

Cell Type	Advantages and Disadvantages	Citations
TILs	Patient specific; difficult to obtain; may be highly cancer specific. TCR is typically low affinity. Cells engineered to express IL-2, IL-7, IL-12, IL-15, and IL-18 show enhanced expansion and function; IL-12 secretion produced toxicity.	[11,12,13,14,15,16,17,18]
TCR engineered T cells	High affinity and specific; patient specific or disease specific. Difficult to make. Mispairing with endogenous TCR and variability of expression are issues.	[19,20]
CAR T cells	Highly effective with B cell neoplasms. Patient specific, expensive, and sometimes toxic. Already an established carrier of many biologics as “armored CARs”^.^	[6,21]
Macrophages	Limited experience to date in secreting drugs. May be difficult to obtain and expand. Can link the innate and adaptive immune response.	[22,23,24]
B cells	Cells are capable of large protein production. May be difficult to obtain and expand.	[25,26,27,28,29,30,31]
CIK and NK cells	May be less toxic than T cells. Less GVHD and CRS may allow allogeneic uses. IL-15 armoring prolongs activity.	[32,33,34]
iPSCs	Off-the-shelf cells are possible. Risks of insertional mutagenesis exist.	[35,36,37,38,39]

**Table 2 cancers-12-02175-t002:** Approaches to transduction of cellular therapeutic agents.

Vector Design or Approach	Advantages and Disadvantages	Citations
Multicistronic with 2A peptide or furin cleavage site	Small size. Allows the production of separate proteins from one promotor, but cistrons cannot be differentially regulated. Varied expression on either side of the 2A element.	[74,75]
Multiple promoters or bicistronic with an IRES site	Large size and often reduced expression of the second product, which may or may not be desired.	[78,79]
Co-transduction	Efficiencies often low and increases cost and complexity.	[80,81,82]
Nanoparticle with DNA or RNA for in vivo use	Allows off-the-shelf engineering as cells manipulated in vivo. Size of constructs and persistence may be limiting. RNA may require multiple doses.	[83,84,85]
Gene editing and transposons.	Allows control of the insertion site, reducing potential adverse effects and controlling expression; reduces TCR misparing. Low efficiency.	[86,87,88,89]
Implanted polymers for in vivo use	Allows off-the-shelf engineering as cells manipulated in vivo. Limited by access to tumors. Long-term effects of implant unknown.	[90,91,92]

**Table 3 cancers-12-02175-t003:** Gated “smart” synthetic cell systems.

Gating System	System Type	Logic Decision Made by Cell (Examples)	Citations
Autonomous Gating Systems	Canonical CAR T cell	If antigen, then activate to kill cells.	[174]
	Multiantigen activation	If multiple antigens, then activate to kill cells.	[175]
	Activation dependent	If activated, then initiate transcription of transgene.	[176]
	SynNotch	If membrane-bound ligand, then initiate transcription of transgene	[166,177,178]
	MESA	If soluble ligand, then initiate transcription of transgene.	[179]
	TME gated	If TME, then do B.	[180,181]
	iCAR	If off-target ligand, then do not kill cell antigen-positive cells.	[182]
Remote-controlled Gating Systems	Kill switch	If drug is present, then end therapy.	[183,184]
	SUPRA CAR and UniCAR	If modular recognition molecule bound to cells, then kill cells.	[185,186]
	Synthetic receptor/ligand pairs	If drug, then do or do not B.	[187,188,189]
	Geography restricted	If localized external stimulus, then do B.	[190,191]

**Table 4 cancers-12-02175-t004:** Comparisons of the local delivery of drugs by TCM vs. systemic drug delivery.

Characteristic	Systemic Drug Delivery	Targeted Cellular Micropharmacy (TCM)
Oral or subcutaneous administration	Yes, for many agents.	No. Generally intravenous.
Off-the-shelf availability; Long-term storage	Yes, for most agents	Generally, not currently, but methods to change this are in development.
Control of doses and schedule	Yes, but dose at target site can be variable	Limited to cases in which gating or prodrugs are used.
Systemic toxicity	Often. Can be severe or fatal.	Promises to create less systemic toxicity for the cell-delivered drug.
Therapeutic index (TI)	Often quite limited, resulting in poor efficacy and systemic toxicity.	Local expansion promises to improve TI. Allows drug secretion at target site only.
“Logic” or feedback driven actions	No.	“Smart” logic gates engineered into some forms.
Persistence in body	Typically, hours to days.	Can be days or months to years.
Reactivation when needed.	No.	Yes.
Cost	Low to high.	Currently extremely high.

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
