# Peer review of "Targeted Cellular Micropharmacies: Cells Engineered for Localized Drug Delivery"

_cancers, 2020, doi:10.3390/cancers12082175_

Round 1

Reviewer 1 Report

Dear Authors

Gardner et al reviews about a wide variety of cellular therapy options and various improvisations on cell based therapies to improve the clinical outcome. In general the review covers most of the cell based therapy areas and written in a comprehensive manner.  The authors should consider to including the following points and incorporate to the review. 

  1. Dendritic cell vaccines are widely tested for various malignancies and diseases, even though there are limited long term clinical benefits, the approach is well documented for safety and clinical response. There are promising outcomes from studies with vaccination with naturally circulating DCs (PMID: 30999964), DC: AML fusion based vaccines (PMID: 27928025) and intratumoral DC vaccination studies ((PMID: 31611878).
  2. There are allogenic tumor cell based vaccines are tested for different malignancies. The GVAX trials, α (1,3) Galactosyltransferase Expressing Allogeneic Tumor Cells in NSLC,  DCP-001- DC prime vaccine for AML are tested in patients with promising results. 
  3. There are reports on the potential benefit of in vitro activate NK cells ( IL12+ IL-15+IL-18) in T cell ALL and similar settings ( PMID:28405496). It will be great to include the details into the NK sections.

Author Response

We thank the reviewer for their comments and have added their suggestions, which we feel have contributed to a more comprehensive manuscript.

  1. The suggested DC vaccine referecnes are now discussed in 4 of the manuscript.
  2. GVAX technology is now discussed in section 6.4 of the manuscript
  3. Cytokine-activated NK cells are now discussed in section 2.3 of the mansucript.

Reviewer 2 Report

The authors summarized the current knowledge of engineered cellular therapies and present a comprehensive and well-structured view of this topic. The review is well written and fits the scope of the Cancers. I am pleased to recommend the acceptance of the paper.  

Author Response

We thank the reviewer for their comments. No changes were requested.

Reviewer 3 Report

In this manuscript, authors reviewed literature associated with critical advances and considerations underway in developing the novel and attractive therapies using human cell-based approaches for the treatment of cancers and other disease. Overall, this review article provides a comprehensive information regarding “Targeted Cellular Micropharmacies (TCM)” as the potential anticancer drugs in the near future. Few questions need to be addressed before publication:

  • It is not clear that any specific types of cancers would be able to use TCM, successfully. On the other hand, what types of cancer patients cannot (or just limited) get benefits from these engineered cellular therapies?
  • Authors might describe that the possible side/adverse effects when patients receive TCM for a long term treatment.
  • Can TCM be used for the treatment of advanced cancer progression, such as relapse, metastasis and/or drug resistance?

Author Response

We thank the reviewer for their insightful comments and have added various discussion points relating to the items that they brought up. We believe that these have strengthened the manuscript by emphasizing important considerations and limitations of TCM technology.

All three points brought up by the reviewer are discussed throughout the introductory portion of section 10.